# Visceral Machines: Risk-Aversion in Reinforcement Learning with Intrinsic Physiological Rewards

**Daniel McDuff and Ashish Kapoor**
Microsoft Research
Redmond, WA
{damcduff,akapoor}@microsoft.com

## Abstract

As people learn to navigate the world, autonomic nervous system (e.g., "fight or flight") responses provide intrinsic feedback about the potential consequence of action choices (e.g., becoming nervous when close to a cliff edge or driving fast around a bend.) Physiological changes are correlated with these biological preparations to protect one-self from danger. We present a novel approach to reinforcement learning that leverages a task-independent intrinsic reward function trained on peripheral pulse measurements that are correlated with human autonomic nervous system responses. Our hypothesis is that such reward functions can circumvent the challenges associated with sparse and skewed rewards in reinforcement learning settings and can help improve sample efficiency. We test this in a simulated driving environment and show that it can increase the speed of learning and reduce the number of collisions during the learning stage.

## 1 Introduction

The human autonomic nervous system (ANS) is composed of two branches. One of these, the sympathetic nervous system (SNS), is "hard-wired" to respond to potentially dangerous situations often reducing, or by-passing, the need for conscious processing. The ability to make rapid decisions and respond to immediate threats is one way of protecting oneself from danger. Whether one is in the African savanna or driving in Boston traffic.

The SNS regulates a range of visceral functions from the cardiovascular system to the adrenal system (Jansen et al., 1995). The anticipatory response in humans to a threatening situation is for the heart rate to increase, heart rate variability to decrease, blood to be diverted from the extremities and the sweat glands to dilate. This is the body's "fight or flight" response.

While the primary role of these anticipatory responses is to help one prepare for action, they also play a part in our appraisal of a situation. The combination of sensory inputs, physiological responses and cognitive evaluation form emotions that influence how humans learn, plan and make decisions (Loewenstein & Lerner, 2003). Intrinsic motivation refers to being moved to act based on the way it makes one feel. For example, it is generally undesirable to be in a situation that causes fear and thus we might choose to take actions that help avoid these types of contexts in future. This is contrasted with extrinsic motivation that involves explicit goals (Chentanez et al., 2005).

Driving is an everyday example of a task in which we commonly rely on both intrinsic and extrinsic motivations and experience significant physiological changes. When traveling in a car at high-speed one may experience a heightened state of arousal. This automatic response is correlated with the body's reaction to the greater threats posed by the situation (e.g., the need to adjust steering more rapidly to avoid a pedestrian that might step into the road). Visceral responses are likely to preempt accidents or other events (e.g., a person will become nervous before losing control and hitting someone). Therefore, these signals potentially offer an advantage as a reward mechanism compared to extrinsic rewards based on events that occur in the environment, such as a collision. This paper provides a reinforcement learning (RL) framework that incorporates reward functions for achieving task-specific goals and also minimizes a cost trained on physiological responses to

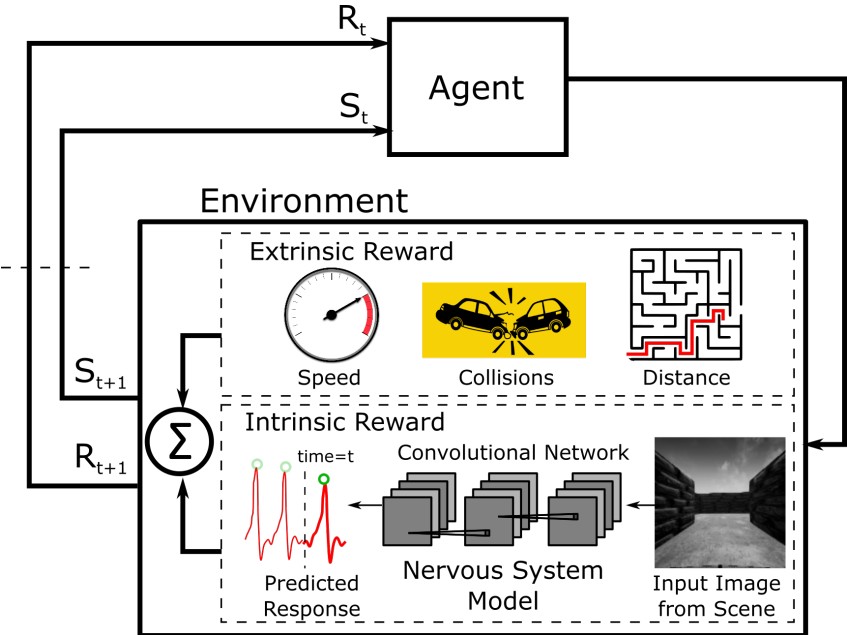

Figure 1: We present a novel approach to reinforcement learning that leverages an artificial network trained on physiological signals correlated with autonomic nervous system responses.

the environment that are correlated with stress. We ask if such a reward function with extrinsic and intrinsic components is useful in a reinforcement learning setting. We test our approach by training a model on real visceral human responses in a driving task.

The key challenges of applying RL in the real-world include the amount of training data required and the high-cost associated with failure cases. For example, when using RL in autonomous driving, rewards are often sparse and skewed. Furthermore, bad actions can lead to states that are both catastrophic and expensive to recover from. While much of the work in RL focuses on mechanisms that are task or goal dependent, it is clear that humans also consider the feedback from the body's nervous system for action selection. For example, increased arousal can help signal imminent danger or failure to achieve a goal. Such mechanisms in an RL agent could help reduce the sample complexity as the rewards are continually available and could signal success or failure before the end of the episode. Furthermore, these visceral signals provide a warning mechanism that in turn could lead to safer explorations.

Our work is most closely related to that in intrinsically motivated learning (Chentanez et al., 2005; Zheng et al., 2018; Haber et al., 2018; Pathak et al., 2017) that uses a combination of intrinsic and extrinsic rewards and shows benefits compared to using extrinsic rewards alone. The key distinction in our work is that we specifically aim to build intrinsic reward mechanisms that are visceral and trained on signals correlated with human affective responses. Our approach could also be considered a form of imitation learning (Ross et al., 2011; Ross & Bagnell, 2014; Ho & Ermon, 2016; Chang et al., 2015) as we use the signal from a human expert for training. However, a difference is that our signal is an implicit response from the driver versus an explicit instruction or action which might commonly be the case in imitation learning.

The structural credit assignment problem, or generalization problem, aims to address the challenge posed by large parameter spaces in RL and the need to give the agent the ability to guess, or have some intuition about new situations based on experience (Lin, 1992). A significant advantage of our proposed method is the reduced sparsity of the reward signal. This makes learning more practical in a large parameter space. We conduct experiments to provide empirical evidence that this can help reduce the number of epochs required in learning. In a sense, the physiological response could be considered as an informed guess about new scenarios before the explicit outcome is known. The challenge with traditional search-based structured prediction is the assumptions that must be made

in the search algorithms that are required (Daumé et al., 2009). By training a classifier using a loss based on the human physiological response this problem can potentially be simplified.

The core contributions of this paper are to: (1) present a novel approach to learning in which the reward function is augmented with a model learned directly from human nervous system responses, (2) show how this model can be incorporated into a reinforcement learning paradigm and (3) report the results of experiments that show the model can improve both safety (reducing the number of mistakes) and efficiency (reducing the sample complexity) of learning.

In summary, we argue that a function trained on physiological responses could be used as an intrinsic reward or value function for artificially intelligent systems, or perhaps more aptly artificially emotionally intelligent systems. We hypothesize that incorporating intrinsic rewards with extrinsic rewards in an RL framework (as shown in Fig 1) will both improve learning efficiency as well as reduce catastrophic failure cases that occur during the training.

## 2 BACKGROUND

### 2.1 SYMPATHETIC NERVOUS SYSTEM

The SNS is activated globally in response to fear and threats. Typically, when threats in an environment are associated with a "fight of flight" response the result is an increase in heart rate and perspiration and release of adrenaline and cortisol into the circulatory system. These physiological changes act to help us physically avoid danger but also play a role in our appraisal of emotions and ultimately our decision-making. A large volume of research has found that purely rational decision-making is sub-optimal (Lerner et al., 2015). This research could be interpreted as indicating that intrinsic rewards (e.g., physiological responses and the appraisal of an emotion) serve a valuable purpose in decision-making. Thus, automatic responses both help people act quickly and in some cases help them make better decisions. While these automatic responses can be prone to mistakes, they are vital for keeping us safe. Logical evaluation of a situation and the threat it presents is also important. Ultimately, a combination of intrinsic emotional rewards and extrinsic rational rewards, based on the goals one has, is likely to lead to optimal results.

### 2.2 REINFORCEMENT LEARNING

We consider the standard RL framework, where an agent interacts with the environment (described by a set of states $\mathcal{S}$), through a set of actions ($\mathcal{A}$). An action $a_t$ at a time-step $t$ leads to a distribution over the possible future state $p(s_{t+1}|s_t, a_t)$, and a reward $r : \mathcal{S} \times \mathcal{A} \to \mathbb{R}$. In addition, we start with a distribution of initial states $p(s_0)$ and the goal of the agent is to maximize the discounted sum of future rewards: $R_t = \sum_{i=t}^{\infty} \gamma^{i-t} r_i$, where $\gamma$ is the discount factor. Algorithms such as Deep Q-Networks (DQN) learn a Neural-Network representation of a deterministic policy $\pi : \mathcal{S} \to \mathcal{A}$ that approximates an optimal Q-function: $Q^*(s, a) = \mathbb{E}_{s' \sim p(\cdot|s,a)}[r(s, a) + \gamma \max_{a' \in \mathcal{A}} Q^*(s', a')]$.

The application of RL techniques to real-world scenarios, such as autonomous driving, is challenging due to the high sample complexity of the methods. High-sample complexity arises due to the credit-assignment problem: it is difficult to identify which specific action from a sequence was responsible for a success or failure. This issue is further exacerbated in scenarios where the rewards are sparse. Reward shaping (Ng et al., 1999; Russell, 1998) is one way to deal with the sample complexity problem, in which heuristics are used to boost the likelihood of determining the responsible action.

We contrast sparse episodic reward signals in RL agents with physiological responses in humans. We conject that the sympathetic nervous system (SNS) responses for a driver are as informative and useful, and provide a more continuous form of feedback. An example of one such SNS response is the volumetric change in blood in the periphery of skin, controlled in part through vasomodulation. We propose to use a reward signal that is trained on a physiological signal that captures sympathetic nervous system activity. The key insight being that physiological responses in humans indicate adverse and risky situations much before the actual end-of-episode event (e.g. an accident) and even if the event never occurs. By utilizing such a reward function, not only is the system able to get a more continuous and dense reward but it also allows us to reason about the credit assignment problem. This is due to the fact that an SNS response is often tied causally to the set of actions responsible for the eventual success or failure of the episode.

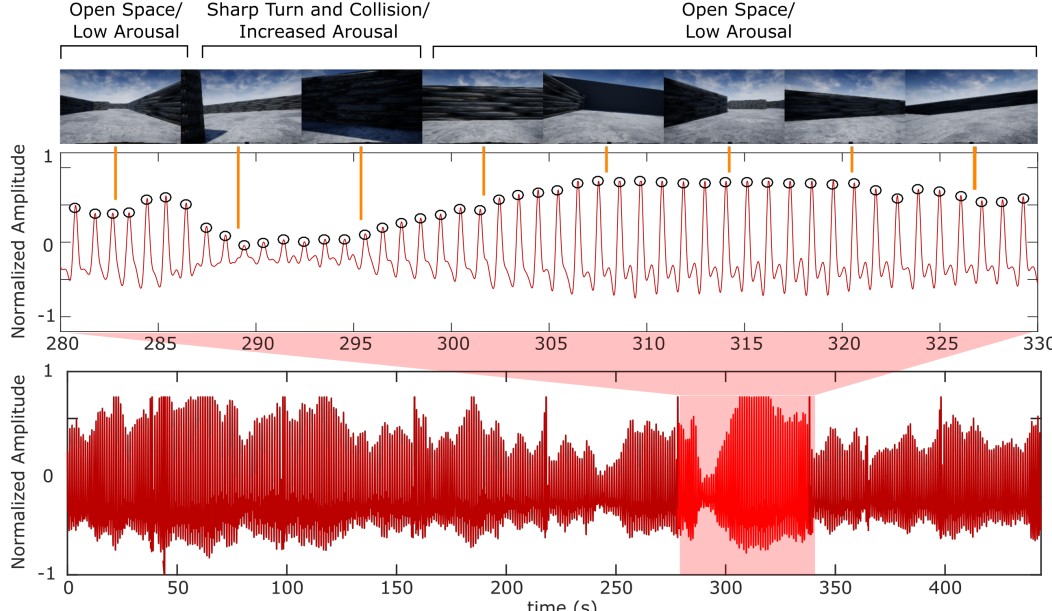

Figure 2: An example of the blood volume pulse wave during driving in the simulated environment. A zoomed in section of the pulse wave with frames from the view of the driver are shown. Note how the pulse wave pinches between seconds 285 and 300, during this period the driver collided with a wall while turning sharply to avoid another obstacle. The pinching begins before the collision occurs as the driver's anticipatory response is activated.

Our work is related, in spirit, to a recent study that used facial expressions as implicit feedback to help train a machine learning systems for image generation (Jaques et al., 2018). The model produced sketches that led to significantly more positive facial expressions when trained with input of smile responses from an independent group. However, this work was based on the idea of Social Learning Theory (Bandura & Walters, 1977) and that humans learn from observing the behaviors of others, rather than using their own nervous system response as a reward function.

## 3  THE PROPOSED FRAMEWORK

Our proposal is to consider a reward function that has both an *extrinsic* component $r$ and an *intrinsic component* $\tilde{r}$. The extrinsic component rewards behaviors that are task specific, whereas the intrinsic component specifically aims to predict a human physiological response to SNS activity and reward actions that lead to states that reduce stress and anxiety. The final reward $\hat{r}$ then is a function that considers both the extrinsic as well as intrinsic components $\hat{r} = f(r, \tilde{r})$. Theoretically, the function $f(\cdot, \cdot)$ can be fairly complex and one possibility would be to parameterize it as a neural network. For simplicity, we consider linear combinations of the extrinsic and intrinsic rewards in this paper. Formally, lets consider an RL framework based on a DQN with reward $r$. We propose to use a modified reward $\hat{r}$ that is a convex combination of the original reward $r$ and a component that mirrors human physiological responses $\tilde{r}$:

$$\hat{r} = \lambda r + (1 - \lambda)\tilde{r} \tag{1}$$

Here $\lambda$ is a weighting parameter that provides a trade-off between the desire for task completion (extrinsic motivation) and physiological response (intrinsic motivation). For example, in an autonomous driving scenario the task dependent reward $r$ can be the velocity, while $\tilde{r}$ can correspond to physiological responses associated with safety. The goal of the system then is to complete the task while minimizing the physiological arousal response. The key challenge now is to build a computational model of the intrinsic reward $\tilde{r}$ given the state of the agent.

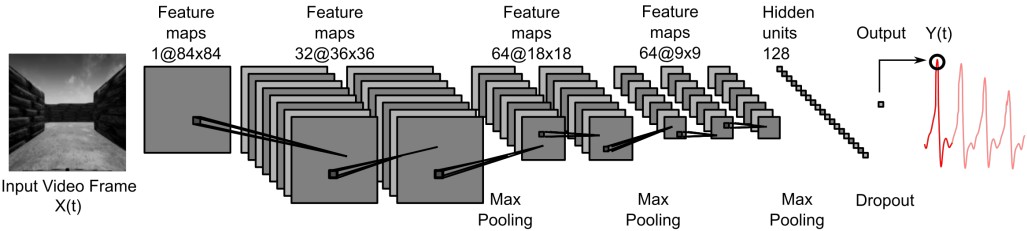

Figure 3: We used an eight-layer CNN (seven convolutional layers and a fully connected layer) to estimate the normalized pulse amplitude derived from the physiological response of the driver. The inputs were the frames from the virtual environment, AirSim.

In the rest of the paper we focus on the autonomous driving scenario as a canonical example and discuss how we can model the appropriate physiological responses and utilize them effectively in this framework. One of the greatest challenges in building a predictive model of SNS responses is the collection of realistic ground truth data. In this work, we use high-fidelity simulations (Shah et al., 2018) to collect physiological responses of humans and then train a deep neural network to predict SNS responses that will ultimately be used during the reinforcement learning process. In particular, we rely on the photoplethysmographic (PPG) signal to capture the volumetric change in blood in the periphery of the skin (Allen, 2007). The blood volume pulse waveform envelope pinches when a person is startled, fearful or anxious, which is the result of the body diverting blood from the extremities to the vital organs and working muscles to prepare them for action, the "fight or flight" response. Use of this phenomenon in affective computing applications is well established and has been leveraged to capture emotional responses in marketing/media testing (Wilson & Sasse, 2000), computer tasks (Scheirer et al., 2002) and many other psychological studies (L. Fredrickson & Levenson, 1998; Gross, 2002). The peripheral pulse can be measured unobtrusively and even without contact (Poh et al., 2010; Chen & McDuff, 2018), making it a good candidate signal for scalable measurement. We leverage the pulse signal to capture aspects of the nervous system response and our core idea is to train an artificial network to mimic the pulse amplitude variations based on the visual input from the perspective of the driver.

To design a reward function based on the nervous system response of the driver in the simulated environment we collected a data set of physiological recordings and synchronized first person video frames from the car. Using this data we trained a convolutional neural network (CNN) to mimic the physiological response based on the input images. Fig 2 shows a section of the recorded blood volume pulse signal with pulse peaks highlighted, notice how the waveform envelope changes.

**Reinforcement Learning Environments:** We performed our experiments in AirSim (Shah et al., 2018) where we instantiated an autonomous car in a maze. The car was equipped with an RGB camera and the goal for the agent was to learn a policy that maps the camera input to a set of controls (discrete set of steering angles). The agent's extrinsic reward can be based on various driving related tasks, such as keeping up the velocity, making progress towards a goal, traveling large distances, and can be penalized heavily for collisions. Fig 2 shows example frames captured from the environment. The maze consisted of walls and ramps and was designed to be non-trivial to navigate for the driver.

**Intrinsic Reward Architecture:** We used a CNN to predict the normalized pulse amplitude derived from the physiological response of the driver. The image frames from the camera sensor in the environment served as an input to the network. The input frames were downsampled to $84 \times 84$ pixels and converted to grayscale format. They were normalized by subtracting the mean pixel value (calculated on the training set). The network architecture is illustrated in Fig 3. A dense layer of 128 hidden units preceded the final layer that had linear activation units and a mean square error (MSE) loss, so the output formed a continuous signal from 0 to 1.

**Training the Reward Network:** We recruited four participants (2 male, 2 female) to drive a vehicle around the maze and to find the exit point. All participants were licensed drivers and had at least seven years driving experience. For each participant we collected approximately 20 minutes ($\sim$24,000 frames at a resolution of $256 \times 144$ pixels and frame rate of 20 frames-per-second) of continuous

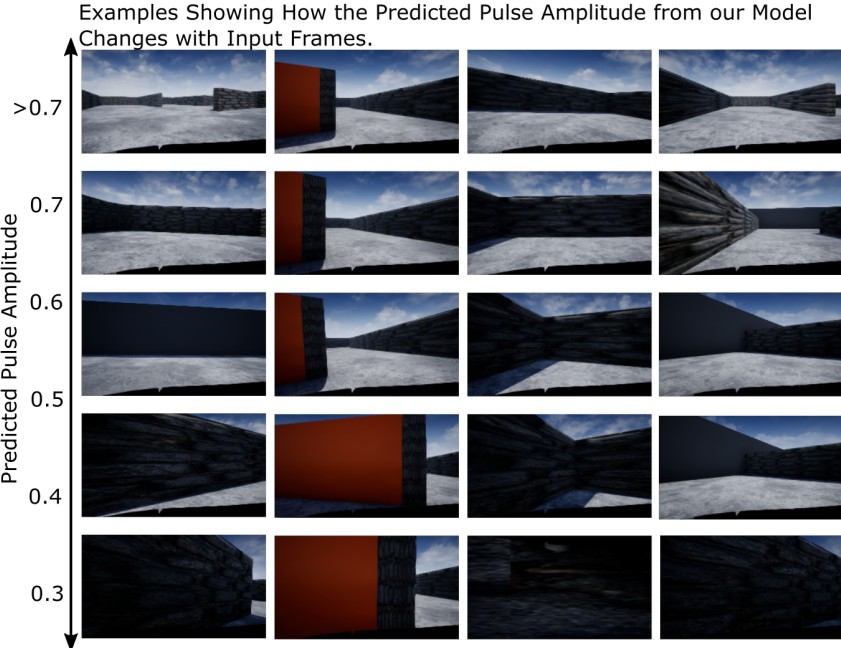

Figure 4: Frames from the environment ordered by the predicted pulse amplitude from our CNN intrinsic reward model. A lower value indicates a higher SNS/"fight or flight" response. This is associated with more dangerous situations (e.g., driving close to walls and turning in tight spaces).

driving in the virtual environment (for a summary of the data see Table 1). In addition, the PPG signal was recorded from the index finger of the non-dominant hand using a Shimmer3[1] GSR+ with an optical pulse sensor. The signal was recorded at 51.6Hz. The physiological signals were synchronized with the frames from the virtual environment using the same computer clock. A standard custom peak detection algorithm (McDuff et al., 2014) was used to recover the systolic peaks from the pulse waveform. The amplitudes of the peaks were normalized, to a range 0 to 1, across the entire recording. Following the experiment the participants reported how stressful they found the task (Not at all, A little, A moderate amount, A lot, A great deal). The participants all reported experiencing some stress during the driving task. The frames and pulse amplitude measures were then used to train the CNN (details in the next section). The output of the resulting trained CNN (the visceral machine) was used as the reward ($\tilde{r}$ = CNN Output) in the proposed framework.

## 4 EXPERIMENTS AND RESULTS

We conducted experiments to answer: (1) if we can build a deep predictive model that estimates a peripheral physiological response associated with SNS activity and (2) if using such predicted responses leads to sample efficiency in the RL framework. We use DQN as a base level approach and build our proposed changes on top of it. We consider three different tasks in the domain of autonomous driving: (a) keeping the velocity high ($r$ is instantaneous velocity), (b) traveling long straight-line distances from the origin ($r$ is absolute distance from origin) without any mishaps and (c) driving towards a goal ($r = 10$ if the goal is achieved). While the velocity and distance task provides dense rewards, the goal directed task is an example where the rewards are sparse and episodic. Note that in all three cases we terminate the episode with a high negative reward ($r = $-10) if a collision happens.

### 4.1 HOW WELL CAN WE PREDICT BVP AMPLITUDE?

We trained five models, one for each of the four participants independently and one for all the participants combined. In each case, the first 75% of frames from the experimental recordings were

---

[1]http://www.shimmersensing.com/

| Part. | Gender | Age (Yrs) | Driving Exp. (Yrs) | Was the Task Stressful? | # Frames | Testing Loss (RMSE) | Testing Loss Improve. over Random (RMSE) |
|---|---|---|---|---|---|---|---|
| P1 | M | 31 | 8 | A lot | 28,968 | .189 | .150 |
| P2 | F | 37 | 20 | A lot | 23,005 | .100 | .270 |
| P3 | F | 33 | 7 | A little | 23,789 | .102 | .234 |
| P4 | M | 31 | 15 | A little | 25,972 | .116 | .194 |
| All P. | | | | | 101,734 | .115 | .210 |

Table 1: Summary of the Data and Testing Loss of our Pulse Amplitude Prediction Algorithm. We Compare the Testing Loss from the CNN Model with a Random Baseline. In All Cases the CNN Gave a Significantly Lower RMSE.

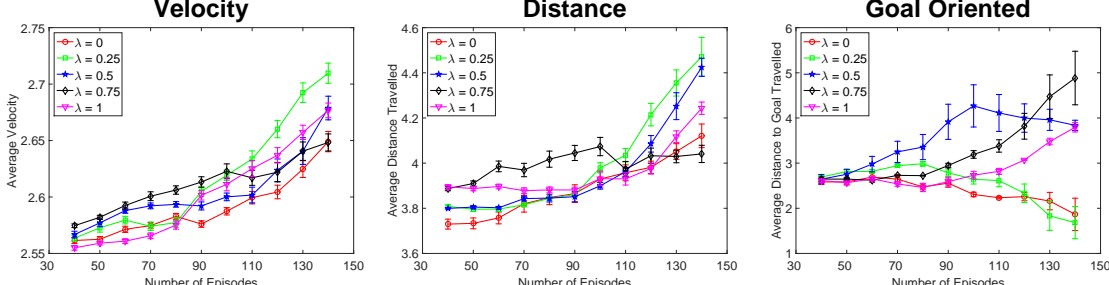

Figure 5: The graph plots average *extrinsic* reward per episode as the system evolves over time for different values of $\lambda$. For all three tasks we observe that using appropriately balanced visceral rewards with the extrinsic reward leads to better learning rates when compared to either vanilla DQN (magenta triangle $\lambda = 1$) or DQN that only has the visceral component (red circle $\lambda = 0$). The error bars in the plots correspond to standard error- non-overlapping bars indicate significant differences ($p<0.05$).

taken as training examples and the latter 25% as testing examples. The data in the training split was randomized and a batch size of 128 examples was used. Max pooling was inserted between layers 2 and 3, layers 4 and 5, and layers 7 and 8. To overcome overfitting, a dropout layer (Srivastava et al., 2014) was added after layer 7 with rate $d_1 = 0.5$. The loss during training of the reward model was the mean squared error. Each model was trained for 50 epochs after which the training root mean squared error (RMSE) loss was under 0.1 for all models. The RMSE was then calculated on the independent test set and was between 0.10 and 0.19 for all participants (see Table 1). As a naive baseline the testing loss for a random prediction was 0.210 greater on average. In all cases the CNN model loss was significantly lower than the random prediction loss (based on unpaired T-Tests). Fig 4 illustrates how a trained CNN associates different rewards to various situations. Specifically, we show different examples of the predicted pulse amplitudes on an independent set of the frames from the simulated environment. A lower value indicates a higher stress response. Quantitatively and qualitatively these results show that we could predict the pulse amplitude and that pinching in the peripheral pulse wave, and increased SNS response, was associated with approaching (but not necessarily contacting) obstacles. The remaining results were calculated using the model from P1; however, similar data were obtained from the other models indicating that the performance generalized across participants.

## 4.2 DOES THE VISCERAL REWARD COMPONENT IMPROVE PERFORMANCE?

We then used the trained CNN as the visceral reward component in a DQN framework and used various values of $\lambda$ to control the relative weight when compared to the task dependent reward component. Fig 5 shows the mean *extrinsic* reward per episode as a function of training time. The plots are averaged over 10 different RL runs and we show plots for different values of $\lambda$. When $\lambda = 1$ that RL agent is executing vanilla DQN, whereas $\lambda = 0$ means that there is no extrinsic reward signal. For all three tasks, we observe that the learning rate improves significantly when $\lambda$ is either non-zero or not equal to 1. The exact value of the optimal $\lambda$ varies from task to task, due to the slightly different final reward structures in the different tasks. One of the main reasons is that the

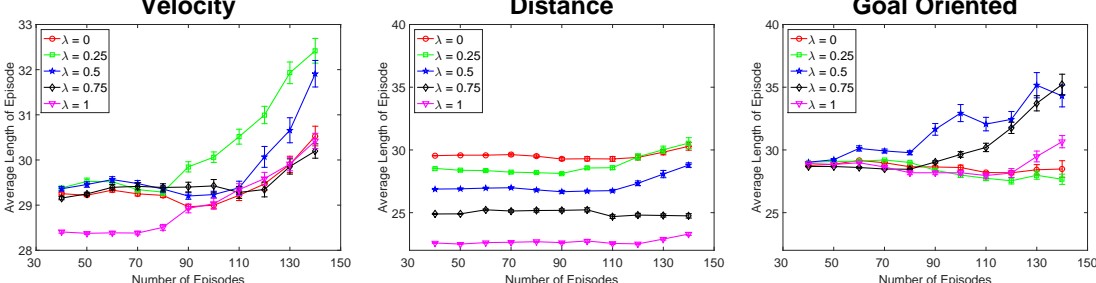

Figure 6: The graph plots average length per episode as the system evolves over time. For all three tasks we observe that using visceral reward components leads to better longer episodes when compared to vanilla DQN ($\lambda = 1$). This implies that the agent with the visceral reward component becomes more cautious about collisions sooner. The error bars in the plots correspond to standard error- non-overlapping bars indicate significant differences ($p < 0.05$).

rewards are non-sparse with the visceral reward component contributing effectively to the learning. Low values of $\lambda$ promote a risk-averse behavior in the agent and higher values $\lambda$ train an agent with better task-specific behavior, but require longer periods of training. It is the mid-range values of $\lambda$ (e.g. 0.25) that lead to optimal behavior both in terms of the learning rate and the desire to accomplish the mission.

### 4.3    DOES THE VISCERAL REWARD COMPONENT HELP REDUCE COLLISIONS?

Fig 6 plots how the average length of an episode changes with training time for different values of $\lambda$. Note that we consider an episode terminated when the agent experiences a collision, so the length of the episode is a surrogate measure of how cautious an agent is. We observed that a low value of $\lambda$ leads to longer episodes sooner while high values do not lead to much improvement overall. Essentially, a low value of $\lambda$ leads to risk aversion without having the desire to accomplish the task. This results in a behavior where the agent is happy to make minimal movements while staying safe. Fig 6 (Distance) shows that the average length of the episodes does not increase with the number of episodes. This is because with increasing numbers of episodes the car travels further but also faster. These two factors cancel one another out resulting in episodes with similar lengths (durations).

### 4.4    WHAT IS THE EFFECT OF A DECAYING VISCERAL REWARD?

What happens if we introduce a time varying intrinsic reward that decays over time? We also ran experiments with varying $\lambda$:

$$\lambda = 1 - \frac{1}{N_{\text{Episode}}} \tag{2}$$

Where, $N_{\text{Episode}}$ is the current episode number. As before, the reward was calculated as in Eqn. 1. Therefore, during the first episode $\lambda$ is equal to zero and the reward is composed entirely of the intrinsic component. As the number of episodes increases the contribution of the intrinsic reward decreases. By episode 95 the intrinsic reward contributes to less than 2% of the total reward. Fig 7 plots the average velocity (left) and average length (right) per episode as the system evolves over time. The blue lines show the performance with a time decaying contribution from the intrinsic reward. These are compared with the best $\lambda$ ($= 0.25$) (red lines) from the previous velocity experiments (see Figs 5 and 6). The episode length is quite superior with the time decaying intrinsic reward. This is because we are directly optimizing for safety initially and the agent quickly learns not to crash. This highlights the value of the intrinsic reward in increasing the safety of the vehicle and extending the length of episodes, especially initially, when the vehicle has little knowledge of how to behave.

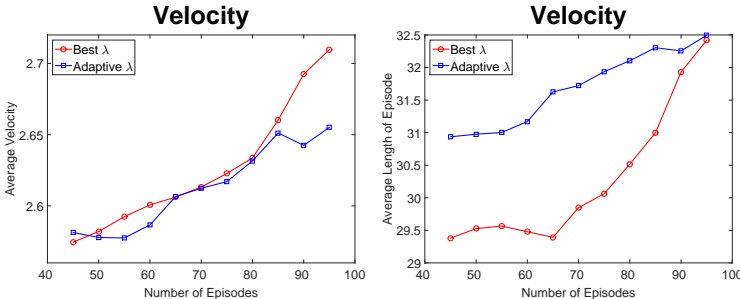

Figure 7: Average velocity (left) and average length (right) per episode as the system evolves over time. Blue) Performance with a time decaying contribution from the intrinsic reward (decaying at 1/(No. of Episodes)). Red) Performance of the best $\lambda$ at each episode (red lines) from the previous velocity experiments (see Fig. 5 and 6). The episode length is superior with the time decaying intrinsic reward because we are directly optimizing for safety initially and the agent quickly learns not to crash.

Figure 8: Comparison of the CNN based intrinsic reward with a reward shaping mechanism. The plots are (left) average extrinsic reward per episode and (right) length of episode as the system evolves and show the advantages of the CNN based approach in both cases.

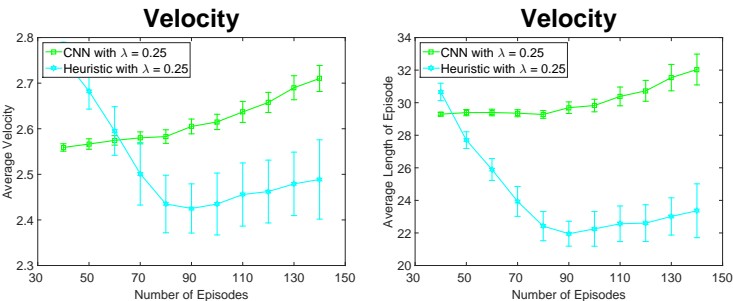

## 4.5 How does the performance compare to reward shaping?

Something we question is, is the CNN predicting the SNS responses doing more than predicting distances to the wall and if there are ways in which the original reward can be shaped to include that information? We did RL experiments where we compared the proposed architecture ($\lambda = 0.25$) with an agent that replaced the intrinsic reward component with the reward $1 - \exp[-|\text{distance to wall}|]$. Note that such distance measures are often available through sensors (such as sonar, radar etc.); however, given the luxury of the simulation we chose to use the exact distance for simplicity. Fig 8 shows both the average reward per episode as well as average length per episode, for the velocity task, as a function of training time. We observed that the agent that had used the CNN for the intrinsic reward component performs better than the heuristic. We believe that the trained CNN is far richer than the simple distance-based measure and is able to capture the context around the task of driving the car in confined spaces (e.g., avoiding turning at high speeds and rolling the car).

## 5 Conclusion and Future Work

Heightened arousal is an key part of the "fight or flight" response we experience when faced with risks to our safety. We have presented a novel reinforcement learning paradigm using an intrinsic reward function trained on peripheral physiological responses and extrinsic rewards based on mission goals. First, we trained a neural architecture to predict a driver's peripheral blood flow modulation based on the first-person video from the vehicle. This architecture acted as the reward in our reinforcement learning step. A major advantage of training a reward on a signal correlated with the sympathetic nervous system responses is that the rewards are non-sparse - the negative reward starts to show up much before the car collides. This leads to efficiency in training and with proper design can lead to policies that are also aligned with the desired mission. While emotions are important for decision-making (Lerner et al., 2015), they can also detrimentally effect decisions in certain contexts. Future work will consider how to balance intrinsic and extrinsic rewards and include extensions to representations that include multiple intrinsic drives (such as hunger, fear and pain).

We must emphasize that in this work we were not attempting to mimic biological processes or model them explicitly. We were using a prediction of the peripheral blood volume pulse as an indicator of situations that are correlated with high arousal.

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
