# OpenReview forum: "Visceral Machines: Risk-Aversion in  Reinforcement Learning with Intrinsic Physiological Rewards"
_ICLR.cc/2019/Conference_

### Official Review · AnonReviewer3 · 2018-11-04
**Nice idea, not sure it generalizes to other tasks**

**Rating:** 7
**Confidence:** 5

**Review:**

The method proposes to use physiological signals to improve performance of reinforcement learning algorithms. By measuring heart pulse amplitude the authors build an intrinsic reward function that is less sparse that the extrinsic one. It helps to be risk averse and allows getting better performances than the vanilla RL algorithm on a car-driving task.

I found the paper well written and the idea is quite nice. I like the idea that risk aversion is processed as a data-driven problem and not as an optimisation problem or using heuristics. I think this general idea could be pushed further in other cases (like encourage fun, surprise, happiness etc. ).

There are some issues with this paper yet. First, modifying the reward function also modifies the optimal policy. In the specific case of car driving, it may not be bad to modify the policy so that it makes passenger less stressed but in general, it is not good. This is why most of works based on intrinsic motivation also schedule the lambda parameter to decrease with time. This is not something explored in this paper. Also, this work is well suited to the car-driving scenario because stress is closely related to risk and accident. But it may not work with other applications. I would thus suggest that the title of the paper reflects the specific case of risk aversion.

---

> ### Author Response · Authors · 2018-11-09
> **Initial response and clarifications.**
>
> Thank you for the review and the suggestion of additional experiments.  We are running experiments regarding the time-varying lambda and will upload the updated manuscript when they are complete in the next few days. We agree that exploring a time varying lambda is of interest. Our goal with this paper was to show that the blood pulse amplitude could be used effectively as an intrinsic reward function that is less sparse than the extrinsic reward from the environment. We feel the experiments capture that and we are happy to include additional results from experiments in which the lambda parameter decreases temporally to illustrate how that influences performance.  The experiments take several days to complete after which we will upload the revised paper.
>
> We are happy to change the title of the paper to reflect the specific case of risk aversion. But we do believe that the principal applies beyond applications in driving to contexts in which stress may be an undesirable state. We propose the following title: “Visceral Machines: Risk-Aversion in Reinforcement Learning with Intrinsic Physiological Rewards”.

---

### Official Review · AnonReviewer1 · 2018-11-06
**An interesting application of RL, but scientifically sloppy**

**Rating:** 6
**Confidence:** 4

**Review:**

Starting from the hypothesis that humans have evolved basic autonomic visceral responses that influence decision making in a meaningful way and that these are at work in driving a car, the authors propose to use such signals within the RL framework. This is accomplished by augmenting the RL reward function with a model learned directly from human nervous system responses. This leads to a
convex combination of extrinsic rewards and visceral responses, with the goal to maximize extrinsic rewards and minimizing the physiological arousal response. The authors first show that they can train a CNN to predict systolic peaks from the pulse waveform based on the input images. The output of this network is then used with parametrically altered weightings in combination with the task related reward to evaluate performance on different driving tasks. The authors show that for different weightings performance on a number of driving tasks performance as measured by the collected extrinsic rewards is better.

Overall, this is an interesting application of RL. It is OK to be inspired by biology, neuroscience, or psychology, but further reaching claims or interpretations of results in these fields need to be chosen carefully. The discussion of neuroscience and psychology are only partially convincing, e.g. there is extensive evidence that autonomic responses are highly dependent on cognition and not just decisions dependent on visceral, autonomic responses of the SNS. Currently, the manuscript is rather loosely switching between inspirations, imprecise claims, and metaphorical implementations with relation to neuroscience. The authors are encouraged to relate their work to some of the multi-criteria and structural credit assignment literature in RL, given the convex combination of rewards.  It may also be important to relate this work to imitation learning, given that the physiological measurements certainly also reflects states and actions by the human agents. While one indication for the reasons of higher extrinsic rewards with the augmented system is mentioned by the authors, namely that the autonomic signal is continuous and while the extrinsic rewards are sparse is convincing, it is not at all clear, why the augmented system performs better as shown in figure 5.

---

> ### Author Response · Authors · 2018-11-09
> **Initial response and clarifications.**
>
> Thank you for your thoughtful review and suggestions of related work. We are running the additional experiments and will upload the updated manuscript when they are complete in the next few days. In the meantime, here is a summary of the changes and responses to your questions. We are tightening up the description of how our system is inspired by biological processes. We want to emphasize that this work is leveraging the peripheral blood volume pulse as a signal that indicates changes in autonomic nervous system arousal, related to stress.  But the system is not mimicking the nervous system or attempting to replicate its processes. We are revising the introduction to make this clear.
>
> We appreciate the comment to relate this work to the imitation learning and credit assignment literature. We are adding to the background sections to tie in this related work. The imitation learning literature is relevant as the physiological reward could be considered similar to feedback from an expert. The structural credit assignment problem, or generalization problem, is related in the way that it helps to address cases in which the parameter space is very large. Our method helps reduce the sparsity of the reward signal and thus makes learning more practical in a large parameter space. We are adding references to the following work on imitation learning and credit assignment:
>
> Search-based Structured Prediction
> Hal Daumé III, John Langford, Daniel Marcu
>
> A Reduction of Imitation Learning and Structured Prediction to No-Regret Online Learning
> Stéphane Ross, Geoffrey J. Gordon, J. Andrew Bagnell
>
> Reinforcement and Imitation Learning via Interactive No-Regret Learning
> Stephane Ross, J. Andrew Bagnell
>
> Generative Adversarial Imitation Learning
> Jonathan Ho, Stefano Ermon
>
> Learning to Search Better Than Your Teacher
> Kai-Wei Chang, Akshay Krishnamurthy, Alekh Agarwal, Hal Daumé III, John Langford
>
> Self-Improving Reactive Agents Based On Reinforcement Learning, Planning and Teaching
> Long-Ji Lin
>
> Regarding explaining why the augmented system performs better, we believe that the sparsity is the best explanation.  The qualitative examples also shed light on the types of situations in which the system is likely to get a high or low reward. Having a less sparse reward that helps the vehicle avoid collisions is beneficial.

---

### Official Review · AnonReviewer2 · 2018-11-06
**Interesting approach, but analysis of the results should be improved**

**Rating:** 6
**Confidence:** 4

**Review:**

Summary:
This submission proposes a reinforcement learning framework based on human emotional reaction in the context of autonomous driving. This relies on defining a reward function as the convex combination of an extrinsic (goal oriented) reward, and an intrinsic reward. This later reward is learnt from experiments with humans performing the task in a virtual environment, for which emotional response is quantified as blood volume pulse wave (BVP). The authors show that including this intrinsic reward lead to a better performance of a deep Q networks, with respect to using the extrinsic reward only.
Evaluation:
Overall the proposed idea is interesting, and the use of human experiments to improve a reinforcement learning algorithm offers interesting perspectives. The weakness of the paper in my opinion is the statistical analysis of the results, the lack of in depth evaluation of the extrinsic reward prediction and the rather poor baseline comparison.
Detailed comments:
1.	Statistical analysis
The significance of the results should be assessed with statistical methods in the following results:
Section 4.1: Please provide and assessment of the significance of the testing loss of the prediction. For example, one could repetitively shuffle blocks of the target time series and quantify the RMSE obtained by the trained algorithm to build an H0 statistic of random prediction.
Section 4.2: the sentence “improves significantly when lambda is either non-zero or not equal to 1” does not seem valid to me and such claim should in any case be properly evaluated statistically (including correction for multiple comparison etc…).
Error bars: please provide a clear description in the figure caption of what the error bars represent. Ideally in case of small samples, box plots would be more appropriate.
2.	Time lags in BVP
It would be interesting to know (from the literature) the typical latency of BVP responses to averse stimuli (and possible the latency of the various mechanisms, e.g. brain response, in the chain from stimuli to BVP). Moreover, as latency is likely a critical factor in anticipating danger before it is too late, it would important to know how the prediction accuracy evolves when learning to predict at different time lags forward in time, and how such level of anticipation influence the performance of the Q-network.
3.	Poor baseline comparison
The comparison to reward shaping in section 4.4 is not very convincing. One can imagine that what counts is not the absolute distance to a wall, but the distance to a wall in the driving direction, within a given solid angle. As a consequence, a better heuristic baseline could be used.
Moreover, it is unclear whether the approaches should be compared with the same lambda: the authors need to provide evidence that the statistics (mean and possibly variance) of the chosen heuristic is match to the original intrinsic reward, otherwise it is obvious that the lambda should be adapted.
4.	Better analysis of figure 5-6(Minor)
I find figure 5-6 very interesting and I would suggest that the authors fully comment on these results. E.g. : (1) why the middle plot of Fig. 6 mostly flat, and why such differences between each curve from the beginning of the training. (2) Why the goal oriented task leads to different optimal lambda, is this just a normalization issue?

---

> ### Author Response · Authors · 2018-11-09
> **Initial response and clarifications.**
>
> Thank you for the detailed comments and constructive suggestions. We are running additional experiments and will upload the updated manuscript when they are complete in the next few days.  We have calculated the baseline RMSE using a random target as suggested by the reviewer. We have added these results and the significance of the difference between the baseline and reported results using an unpaired T-Test. The RMSE of the model predictions is significantly lower for all participants than the RMSE with the random target. The RMSE of the model predictions was 0.21 lower on average. These results have been added to Table 1.
>
> The error bars in the plots correspond to standard error. Non-overlapping error bars correspond to 84% confidence according to z-test (Please see: https://tminka.github.io/papers/minka-errorbars.pdf). Scaling of these by c=1.64, leads to 95% significance when the bars don’t overlap. We will add this information to the figure captions as well as the text in the paper. This should clarify which of the comparisons are significant. We will also modify the claim in the text to correspond to this.
>
> We agree that there is a time delay between a stimulus and the physiological response to that stimulus. Our system was trained to mimic the physiological response of a person, this delay is already modeled into the prediction .  While it might be interesting to introduce an artificial time delay (i.e., to reflect a response that is faster or slower that was actually experienced), we believe our experiments are the most representative of the delays experienced. Please do clarify if we misunderstood your comment.
>
> In this work we were trying to mimic a self-driving car in which the sensors are only facing a single direction and thus distance to the wall was an appropriate assumption.  We are not sure what the reviewer meant in the comment about the statistics of the “chosen heuristic matching to the original intrinsic reward.” We believe that a comparison with the same lambda values is the best initial test to run, without additional reasons to adjust the value of lambda on a case-by-case basis.
>
> Fig. 6 (Distance) shows that the average length of the episodes does not increase with the number of episodes.  This is because with increasing numbers of episodes the car travels further but also faster. These two factors cancel one another out resulting in episodes with similar duration. The goal-oriented task leads to a different optimal lambda due to the different nature of the reward compared to the velocity or distance task.  However, it should be noted that consistently a lambda that is non-zeros and not equal to 1 is optimal.  We are adding more discussion of these results.

---

### Author Response · Authors · 2018-11-13
**Revision uploaded.**

We would like to thank the reviewers again for their constructive and insightful comments.  We have uploaded a revised version of our manuscript with the changes described below in the "initial response and clarifications".  We highlight that we have added additional related work, the experiments and results with a time varying lambda, clarified the error bars and significant of the results and added comparisons with a baseline pulse amplitude prediction.

---

### Meta-Review · Area_Chair1 · 2018-12-16
**An original approach that has significant potential to influence future work in the space of human-robotic interactions.**

**Confidence:** 4
**Recommendation:** Accept (Poster)

**Metareview:**

The paper considers the problem of incorporating human physiological feedback into an autonomous driving system, where minimization of a predicted arousal response is used as an additional source of reward signal, with the intuition that this could be used as a proxy for training a policy that is risk-averse.

Reviewers were generally positive about the novelty and relevance of the approach but had methodological concerns. In particular, concerns about the weighting of the intrinsic vs. extrinsic reward (why under different settings the optimal tradeoff parameter was different, how this affects the optimal policy if the influence of the intrinsic reward is not decreased with time). Additional baseline experiments were requested and performed, and the paper was modified to significantly incorporate other feedback such as drawing connections to imitation learning. A title change was proposed and accepted to reflect the focus on the application of risk aversion (I'd ask that the authors update the paper OpenReview metadata to reflect this).

At a high level, I believe this is an original and interesting contribution to the literature. I have not heard from two of three reviewers regarding whether their concerns were addressed, but given that their concerns appear to me to have been addressed (and their initial scores indicated that the work met the bar for acceptance, if only marginally), I am inclined to recommend acceptance.